# Effects of Steel-Slag Components on Interfacial-Reaction Characteristics of Permeable Steel-Slag–Bitumen Mixture

**DOI:** 10.3390/ma13173885

**Published:** 2020-09-02

**Authors:** Wenhuan Liu, Hui Li, Huimei Zhu, Pinjing Xu

**Affiliations:** College of Materials Science and Engineering, Xi’an University of Architecture and Technology, Xi’an 710055, China; zhuhuimeitj@163.com (H.Z.); xpj6100@163.com (P.X.)

**Keywords:** permeable steel-slag–bitumen mixture, steel-slag–bitumen interface, phase angle, complex shear modulus, high-temperature rutting factor

## Abstract

In this paper, a permeable steel-slag–bitumen mixture (PSSBM) was first prepared according to the designed mixture ratio. Then, the interaction characteristics between steel slag and bitumen were studied. The chemical interaction between bitumen and steel slag was explored with a Fourier-transform infrared spectrometer (FT-IR). The influence of steel-slag chemistry, mineral composition, and bitumen reaction on phase angle, complex shear modulus (CSM), and rutting factor was explored with dynamic shear rheological (DSR) tests. The PSSBM had better properties, including high permeability, water stability, Marshall stability, high-temperature (HT) stability, and low volume-expansion rate. Bitumen-coated steel slag can prevent heavy-metal ions from leaching. In the infrared spectra of the mixture of a chemical component of steel slag (calcium oxide) and bitumen, a new absorption peak at 3645 cm^−1^ was ascribed to the SiO–H stretching vibration, indicating that new organic silicon compounds were produced in the chemical reaction between calcium oxide and bitumen. SiO–H had an obvious enhancement effect on the interfacial adhesion and high-temperature rheological property of the mixture. In the mineral components of steel slag, dicalcium and tricalcium silicate reacted with bitumen and generated new substances. Chemical reactions between tricalcium silicate and bitumen were significant and had obvious enhancement effects on interfacial adhesion and high-temperature rheological properties of the mixture. The results of FT-IR and DSR were basically consistent, which revealed the chemical-reaction mechanism between steel-slag microcomponents and bitumen at the interface. SEM results showed that pits and grooves on the surface of the steel-slag aggregate, and the textural characteristics provide a framework-like function, thus strengthening the strength and adhesion of the steel-slag–bitumen aggregate interface.

## 1. Introduction

Due to the shortage of natural stone and the increase in manufacturing costs of high-quality aggregates in China, the comprehensive utilization of metallurgical steel slag in preparing aggregates has been widely considered. Steel slag is a porous alkaline aggregate, so it has good adhesion to bitumen. In addition, steel slag has high wear resistance and is suitable pavement material [1,2,3,4,5,6,7,8,9,10]. If steel slag can be utilized as an aggregate to prepare permeable steel-slag–bitumen concrete for pavement engineering, it can speed up the construction of ecological pavements, and enhance the resource utilization of steel slag, partially solving the shortage problem of the lack of high-quality natural-stone aggregate [11,12]. Asi et al. [13,14] replaced coarse limestone aggregate of 4.75 mm and above with steel slag, and explored the indirect tensile modulus, rutting resistance, fatigue life, creep modulus, and antistripping performance of dense-graded bitumen mixtures with different contents of coarse aggregate. Bessa et al. [15] studied the abrasion resistance of aggregates, and found that steel slag had better angularity and aggregate surface texture, and stronger abrasion resistance than those of granite and gneiss. Kavussi et al. [16] used electric-furnace slag to substitute the coarse aggregate of 2.36 mm in a limestone–bitumen mixture, and found that the crack propagation of the mixture was largely prevented due to the strong skeleton embedding and extruding effect of steel slag.

If the interface interaction between bitumen and aggregate is weak, the bitumen–aggregate interface is easily damaged, thus leading to early pavement damage. Therefore, in order to solve problems such as particles dropping and peeling off of bitumen pavement, it is necessary to explore the interfacial-interaction characteristics of bitumen and steel-slag aggregate. Interfacial interactions between bitumen and steel slag aggregate can be divided into physical adsorption and chemical reaction. Physical adsorption mainly comes from mechanical adhesion and polar force between aggregate molecules and bitumen molecules, and is a weak interaction. If only physical adsorption exists, the mixture is easily stripped by water. Through adhesion tests, Podol et al. [17] demonstrated that bitumen peeling from the aggregate surface mainly occurred at the aggregate interface. Amit Bhasin et al. [18] measured the surface-area and -energy parameters of aggregates, and found that these parameters determined aggregate–bitumen adhesion. Chen et al. [19] used X-ray photoelectron spectroscopy and scanning electron microscopy (SEM) to analyze the influence of bitumen components on the bitumen–aggregate interface phase.

Chemical reactions between molecules of interface substances mainly involve electrostatic and chemical-bond forces (ionic, covalent, and coordination bonds). When chemical reaction occurs at the interface, the adhesion between bitumen and aggregate is relatively strong. When carbonate or alkaline rock is bonded with bitumen containing sufficient active acidic surface substances, the chemical-adsorption process occurs, and water-insoluble compounds are formed on the contact surface of bitumen and aggregate, so the bitumen layer formed on the surface of aggregate has higher water-damage resistance. Alkaline aggregates have the high adhesion to bitumen containing acidic groups [20,21,22].

At present, the mixing-proportion design and performance of durable bitumen concrete fabricated from steel-slag aggregates has seldom been explored, and the chemical-reaction characteristics of steel-slag–bitumen interface are not fully clarified. In order to give full play to the advantages of steel-slag aggregate, it is necessary to deeply explore the chemical-reaction characteristics of steel-slag–bitumen aggregate interface. In this work, permeable steel-slag–bitumen mixtures were prepared with steel-slag aggregate ground by a vertical mill. The permeability, water stability, Marshall stability, and high-temperature (HT) stability and expansibility of the mixture were tested. X-ray fluorescence spectrometer (XRF) was employed to test chemical composition, and X-ray diffraction (XRD) was used to analyze mineral composition. The dominant mineral and chemical components of steel slag were respectively mixed with bitumen to prepare the mixtures. Dynamic shear rheological (DSR) tests were employed to test the phase angle, complex shear modulus (CSM), and rutting factor of the bitumen mixtures. SEM was employed to test the microscopic morphology of the steel-slag–bitumen interface [23,24,25]. The chemical-reaction characteristics of the steel-slag–bitumen interface were explored from the perspective of the chemical and mineral components of steel slag.

## 2. Materials and Methods

### 2.1. Materials and Reagents

Steel-slag aggregate and micropowder (A Steel Group Co., Ltd., Xi’an, China) with different particle sizes of 2.36–4.75, 4.75–9.5, 9.5–13.2, and 13.2–16 mm used in the test were obtained by high-pressure grinding roller, and had rough surfaces without a weathering phenomenon. Other materials used in the study were A-90 road petroleum bitumen (An Energy Co., Ltd., Jinan, China), alumina, calcium oxide, ferric oxide, silica (A Technology Co., Ltd., Beijing, China), ferroferric oxide, dicalcium silicate, calcium silicate, RO phase (self-made, RO phase is a broad solid solution formed by melting FeO, MgO, and other divalent metal oxides such as MnO), high modulus agent (A Construction Material Co., Ltd., Zhengzhou, China), and deionized water.

Chemical characteristics, XRD test results of metallurgical steel slag, and road-performance index are respectively presented in Table 1 and Table 2 and Figure 1. As shown in Table 1, the dominant chemical components were SiO_2_, Al_2_O_3_, CaO, Fe_2_O_3_, and a certain amount of *f*-CaO. Table 2 shows that all indices were close to and even higher than the standard values. As shown in Figure 1, the major mineral components of steel slag were calcium hydroxide, RO phase dicalcium silicate, and tricalcium silicate, which made the steel slag highly alkaline and able to react with the acid anhydride in petroleum bitumen, thereby promoting the bonding force between steel slag and petroleum bitumen.

The rubber bitumen used in this paper was prepared by adding 20% rubber powder (0.6 mm, 30 mesh) and 0.5% PA-1 antistripping agent to SK 90# bitumen in Korea. Rubber bitumen has high viscosity and a good elastic-recovery effect, which can make the energy generated by fatigue load dissipate more in the form of heat energy, and can effectively avoid fatigue failure caused by cumulative pavement damage. At the same time, 0.3% antirutting agent (based on the total mass of the bitumen mixture) was added to consider high-temperature antirutting performance. As shown in Table 3, all indicators of this bitumen met the requirements of the Technical Specifications for Permeable Bitumen Pavement (CJJT 190–2012).

### 2.2. Sample Preparation and Test Methods

The permeable steel-slag–bitumen mixture (PSSBM) was prepared according to the following procedure. At first, with the effective thickness of bitumen film and specific surface area of the aggregate, the bitumen–aggregate ratio of the mixture was determined to be 6.76%. Then, 362.50 g of steel-slag aggregate of 2.36–4.75 mm, 604.16 g of steel slag aggregate of 4.75–9.5 mm, 241.66 g of steel slag aggregate of 9.5–13.2 mm, and 105 g of steel slag micropowder filler were mixed, placed into an oven, dried to a fixed weight at 170 °C, and then transferred into an E10-H mixer. Lastly, the high modulus agent (4.19 g) and the modified bitumen (87.91 g) were evenly mixed for 3 min.

Toxic leaching of steel slag and permeable steel-slag–bitumen mixture: Samples of steel slag and PSSBM were ground, 10 g was weighed and put into a polyethylene bottle, then 200 mL of toxicity characteristic leaching procedure (TCLP) extractant (pH = 2.88 acetic acid solution) was added, and then turned over and oscillated for 18 h at room temperature. Lastly, the leachate was filtered on 0.45 μ medium-speed filter paper, and the filtrate was nitrated with HNO_3_ to pH < 2. The concentration of metal ions in the filtrate was analyzed by inductively coupled plasma emission spectrometer.

The dynamic shear rheological test was performed as follows. Phase angle, CSM, and rutting factor of PSSBM were analyzed with DSR under conditions of 12% strain and the scanning frequency of 0.1–10 Hz. The gap width and diameter of the DSR samples were, respectively, 1.00 ± 0.05 and 25 mm, and test temperatures were 45 and 55 °C. Bitumen was respectively mixed with four oxide powders (SiO_2_, CaO, Fe_2_O_3_, and Al_2_O_3_) and three mineral components (C_2_S, C_3_S, and RO) according to the mass ratio of 2.5:1 to prepare samples. The modified bitumen sample was heated to 140 °C and maintained for about 60 min. The oxides and mineral-phase powders were dried to a fixed weight at 105 °C and kept at 150 °C for 30 min. Then, oxide and mineral-phase powder samples were mixed for 10 min at 140 °C until no bubbles were observed on the interface of the mixture in order to obtain a uniform mixture.

### 2.3. Performance Test and Characterization

In this paper, PSSBM properties, such as permeability, water stability, Marshall stability, and HT stability and expansibility, were tested on the basis of CJJ/T 190-2012 [26]. Chemical composition, mineral composition, reactions between bitumen and steel slag, microscopic morphology, and the change rule of the phase angle of the mixture of modified bitumen and mineral components of steel slag were obtained with an S4 PIONEER X-ray fluorescence analyzer (Bruker Corporation, Karlsruhe, Baden-Württemberg, Germany), D–MAX/2500 X-ray diffractometer (Rigaku Corporation, Zhaodao, Tokyo, Japan), Fourier infrared spectrometer (PERKINELMER Corporation, Waltham, MA, USA), Quanta 2000B scanning electron microscope (FEI Corporation, Hillsboro, OR, USA), and AR–2000 DSR (TA Instruments Corporation, New Castle, DE, USA), respectively.

## 3. Test Results and Discussion

### 3.1. PSSBM Performance Analysis

The above performance indicators of the mixture are presented in Table 4. The permeability coefficient, Marshall stability, flow value, and residual stability of PSSBM were, respectively, 55.56 mL/s, 9.41 kN, 2.56 mm, and 91.18%, indicating that the mixture had good permeability, water stability, and Marshall stability. The dynamic stability, freeze–thaw splitting-strength ratio, and the volume-expansion rate of PSSBM were 6486, 90%, and 0.45%, indicating that the mixture had high HT stability and low volume-expansion rate.

In this paper, the national first-class surface-water environmental-quality standard was used to judge whether the leaching concentration of heavy-metal ions in steel slag and the PSSBM exceeded the standard. The leaching results of heavy-metal ions from raw steel-slag materials and PSSBM are shown in Table 5 and Figure 2.

Results showed that Cr leaching was not detected in the leaching solution of heavy metals from raw steel-slag materials, but the leaching concentrations of Cu, V, Zn, As, and Mn all exceeded the environmental-quality standards of Class I surface water, so steel slag could not be directly stacked without treatment. Results showed that the leaching concentration of heavy-metal ions was greatly reduced after the steel slag was coated with modified bitumen. V and Cr were detected in the heavy-metal leaching solution of the permeable steel-slag–bitumen mixture, and leaching concentrations of Cu, Zn, As, and Mn were all lower than those of Class I surface-water environmental-quality standards. This is mainly related to the physical curing process. Modified bitumen is an inert material with strong cohesive force. Modified bitumen was wrapped on the surface of steel slag to form a thin film, which separated the steel slag from the leaching solution, thus hindering the leaching of harmful metal ions. In addition, compared with the modified bitumen concrete, in the leaching process, the contact area between mixture and leaching agent was larger, and metal ions are easier to leach. If the leaching concentration of metal ions in bulk materials is lower than that of the Class I surface-water standard, this can ensure that the leaching concentration of heavy-metal ions in the whole permeable steel-slag–bitumen concrete pavement was also lower than this standard.

### 3.2. Chemical-Reaction Characteristics of Steel-Slag Oxides and Modified Bitumen

#### 3.2.1. Oxide Phase Angle—Modified Bitumen Mixtures

Bitumen is a viscoelastic material. Phase angle δ is a lag angle caused by the asynchronous response between sinusoidal stress input to the material and sinusoidal strain generated by the viscous component of the material. For a pure viscous fluid, δ is π/2. For a pure elastic material, δ is 0. For most viscoelastic materials, 0 < δ < π/2. Therefore, the phase angle reflects the viscoelastic proportion of the bitumen material. According to Hashin’s research [27], the phase angle of the composite material without the occurrence of chemical reaction should theoretically be equal to that of the matrix material, but the phase angle of the composites with a chemical reaction was less than that of the matrix material.

Figure 3 shows the phase angles of the oxide-modified bitumen mixtures at 45 and 55 °C. In the temperature range between 45 and 55 °C, the phase angles of the reference sample of modified bitumen, SiO_2_-modified bitumen, Al_2_O_3_-modified bitumen, Fe_2_O_3_-modified bitumen, and CaO-modified bitumen were, respectively, 76.39–87.75°, 76.67–87.28°, 76.24–86.93°, 76.67–86.66°, and 73.45–86.66°. Compared to the phase angle of the modified bitumen sample, the phase angle of the modified bitumen mixture with CaO decreased the most significantly, but the phase angles of the modified bitumen mixture with oxides such as SiO_2_, Al_2_O_3_, and Fe_2_O_3_ rarely changed. Therefore, the chemical reaction between CaO and modified bitumen occurred at 45 and 55 °C, but the chemical reactions between the three other oxides (SiO_2_, Al_2_O_3_, and Fe_2_O_3_) and modified bitumen were not significant.

#### 3.2.2. Complex Shear Modulus of Oxide-Modified Bitumen Mixtures

Figure 4 shows the CSM of oxide-modified bitumen mixtures at various temperatures in the frequency range from 0.1 to 10 Hz. The complicated shear modulus of the modified bitumen and four types of oxide-modified bitumen mixtures increased with the increase in frequency at 45 °C (Figure 4a). At 10 Hz, the CSM of modified bitumen, SiO_2_-modified bitumen mixture, Al_2_O_3_-modified bitumen mixture, CaO-modified bitumen mixture, and Fe_2_O_3_-modified bitumen mixture reached 6.9 × 10^4^, 5.9 × 10^4^, 1.3 × 10^5^, 2.0 × 10^5^, and 9.4 × 10^4^ Pa, respectively. At 45 °C, CSM values of the modified bitumen and four types of oxide-modified bitumen mixtures were decreased in the following order: CaO-modified bitumen mixture > Al_2_O_3_-modified bitumen mixture > Fe_2_O_3_-modified bitumen mixture > SiO_2_-modified bitumen mixture > modified bitumen. Except for the SiO_2_ component, the CSM values of the three other types of oxide–bitumen mixtures were significantly larger than that of the bitumen reference sample, and CaO had the best reinforcing effect. The CSM of modified bitumen and four types of oxide-modified bitumen mixtures increased with the increase in frequency at 55 °C (Figure 4b). At 10 Hz, the CSM of modified bitumen, SiO_2_-modified bitumen mixture, Al_2_O_3_-modified bitumen mixture, CaO-modified bitumen mixture, and Fe_2_O_3_-modified bitumen mixture reached 1.32 × 10^4^, 2.18 × 10^4^, 3.46 × 10^5^, 6.79 × 10^5^, and 2.52 × 10^4^ Pa, respectively. CSM values of the modified bitumen and four types of oxide-modified bitumen mixtures were decreased in the following order: CaO-modified bitumen > Al_2_O_3_-modified bitumen > Fe_2_O_3_-modified bitumen > SiO_2_-modified bitumen > modified bitumen. CSM values of SiO_2_, CaO, Fe_2_O_3_, Al_2_O_3_, and four other oxide–bitumen mixtures at 55 °C were significantly greater than that of the modified bitumen, and the reinforcement effect of CaO was the best.

Figure 5 shows the effect of temperature on the complex shear modulus of oxide component–bitumen mixtures. At 45 and 55 °C, the complex shear modulus of the mixture of bitumen and oxide components such as SiO_2_, CaO, Fe_2_O_3_, and Al_2_O_3_ decreased with the increase in temperature in a frequency range from 0.1 to 10 Hz.

#### 3.2.3. Rheological Properties of Oxide-Modified Bitumen Mixtures at High Temperatures

The DSR was employed to record and present the rheological characters of bitumen binder at a fixed temperature and with a specific frequency in the SHRP (Strategic Highway Research Program) research program of the United States. In Superpave research, the G*/sinδ factor was employed to assess the permanent deformation resistance of bitumen and bitumen mixture composites. The larger the G*/sinδ factor was, the stronger the HT deformation resistance was. Therefore, this was named the rutting factor. In this paper, G*/sinδ was utilized to assess the rheological properties of oxide–bitumen mixtures at high temperatures. In the SHRP research program, the original bitumen at the specified angular velocity (ω = 10 rad/s, equivalent to 1.592 Hz) and shear rate γ = 1.2% were required to meet inequality G*/sinδ ≥ 1.0 kPa.

Figure 6 shows the rutting factors of oxide-modified bitumen mixtures at different temperatures. Figure 6a shows the rutting factors of modified bitumen, and the mixtures of modified bitumen and four oxides (SiO_2_, CaO, Fe_2_O_3_, and Al_2_O_3_) at different frequencies at 45 °C. At 1.592 Hz, the rutting factor of the modified bitumen was about 25 kPa, which met the SHRP requirements. Compared with bitumen, except for SiO_2_-modified bitumen mixture, the three other kinds of oxide-modified bitumen mixtures showed improved rutting factors, indicating that the HT performance of bitumen mixtures mixed with CaO, Al_2_O_3_, and Fe_2_O_3_ had been improved to different degrees. At 10 Hz, the rutting factor of CaO-modified bitumen mixture, SiO_2_-modified bitumen mixture, Al_2_O_3_-modified bitumen mixture, and Fe_2_O_3_-modified bitumen mixture was 3.95, 1.15, 2.03, and 1.59 times that of modified bitumen, respectively. The rutting factors of four different oxide–bitumen mixtures were decreased in the following order: CaO-modified bitumen mixture > Al_2_O_3_-modified bitumen mixture > Fe_2_O_3_-modified bitumen mixture > SiO_2_-modified bitumen mixture.

Figure 6b shows the rutting factors of bitumen samples, and the mixtures of bitumen and four oxides (SiO_2_, CaO, Fe_2_O_3_, and Al_2_O_3_) at different frequencies at 55 °C. At 1.592 Hz, the rutting factor of the modified bitumen was about 2.2 kPa, which met the SHRP requirements. Compared with modified bitumen, except for SiO_2_-modified bitumen mixture, three other kinds of oxide-modified bitumen mixtures had improved rutting factors, indicating that the HT performance of modified bitumen mixtures mixed with CaO, Al_2_O_3_, and Fe_2_O_3_ had been improved to different degrees. The rutting factors of CaO-modified bitumen mixture, SiO_2_-modified bitumen mixture, Al_2_O_3_-modified bitumen mixture, and Fe_2_O_3_-modified bitumen mixture were 4.02, 1.36, 1.87, and 1.5 times that of the bitumen sample, respectively. Therefore, CaO greatly improved the high-temperature rutting-deformation resistance of the steel-slag–bitumen mixture, thus increasing the service life of bitumen pavement. The rutting factors of the mixtures of bitumen and four oxides were decreased in the following order: CaO-modified bitumen mixture > Al_2_O_3_-modified bitumen mixture > Fe_2_O_3_-modified bitumen mixture > SiO_2_-modified bitumen mixture.

Figure 7 shows the effects of temperature on the rutting factor of oxide-component–bitumen mixtures. At 45 and 55 °C, the rutting factor of the mixtures of modified bitumen and oxide components (SiO_2_, CaO, Fe_2_O_3_, and Al_2_O_3_) at 0.1 to 10 Hz decreased with the increase in temperature.

#### 3.2.4. Fourier-Transform Infrared-Spectrum Analysis

Figure 8 shows the FT-IR of the mixtures of modified bitumen and chemical components of steel slag. Most absorption peaks in the infrared spectra of the three mixtures of modified bitumen and three chemical components of steel slag (Al_2_O_3_, Fe_2_O_3_, and SiO_2_) were the superposition consequence of the absorption peaks of modified bitumen and each component, indicating that no chemical reaction occurred between the three components and modified bitumen. A new absorption peak at 3645 cm^−1^ that was observed in the infrared spectra of the CaO–bitumen mixture could be ascribed to SiO–H stretching vibration, indicating that CaO reacted with the modified bitumen to form a new substance, which may have been an organosilicon compound, as inferred from the characteristic peak.

### 3.3. Chemical-Reaction Characteristics between Mineral Components of Steel Slag and Modified Bitumen

#### 3.3.1. Phase Angles of Mixtures of Modified Bitumen–Mineral Components of Steel Slag

The phase angles of the C_2_S-modified bitumen mixture, C_3_S-modified bitumen mixture, and RO phase-modified bitumen mixture were basically the same to that of the modified bitumen at 45 °C (Figure 9a), suggesting that the reaction between the C_2_S, C_3_S, RO phase, and modified bitumen was not significant at 45 °C. The phase angles of the C_2_S-modified bitumen mixture and RO phase-modified bitumen mixture were not significantly smaller than that of the bitumen sample at 55 °C (Figure 9b).

The phase angle of the C_3_S-bitumen mixture was not significantly smaller than that of the modified bitumen at 55 °C, indicating that the chemical reaction between C_3_S and modified bitumen could be significant at 55 °C. Figure 9 shows that, in the temperature range from 45 to 55 °C, the phase-angle ranges of the modified bitumen, C_2_S-modified bitumen mixture, RO phase-modified bitumen mixture, and C_3_S-modified bitumen mixture were 76.39–87.75°, 75.75–88.05°, 76.30–88.08°, and 70.61–86.95°, respectively. After adding C_3_S, the phase angle of the C_3_S-modified bitumen mixture was changed the most significantly, followed by C_2_S-modified bitumen mixture and the RO phase-modified bitumen mixture. The difference indicated that the chemical reaction between C_3_S and modified bitumen was significant, but that the chemical reaction between C_2_S, the RO phase, and modified bitumen was not significant.

#### 3.3.2. Complex Shear Modulus of Mixtures of Bitumen–Mineral Components of Steel Slag

Figure 10 presents the CSM of modified bitumen and the mixtures of bitumen–mineral components of steel slag at different temperatures. The CSM of modified bitumen and the three mixtures of bitumen–mineral components of steel slag increased with the increase in frequency at 45 °C (Figure 10a). At 10 Hz, the CSM of modified bitumen, C_3_S-modified bitumen mixture, C_2_S-modified bitumen mixture, and RO phase-modified bitumen reached 6.9 × 10^4^, 1.12 × 10^5^, 8.5 × 10^4^, and 6.79 × 10^4^ Pa, respectively. The CSM values of the modified bitumen and the three mixtures of bitumen–mineral components of steel slag decreased in the following order: C_3_S-modified bitumen > C_2_S-modified bitumen > RO phase-modified bitumen > modified bitumen. The difference indicated that C_2_S and C_3_S could obviously increase the CSM of modified bitumen mixture, and C_3_S had the best reinforcement effect.

The CSM of modified bitumen and the mixtures of bitumen–mineral components of steel slag (C_2_S, C_3_S, and RO) at 55 °C in the frequency range of 0.1 to 10 Hz is presented in Figure 10b. The CSM of modified bitumen samples and three mixtures of modified bitumen–mineral components of steel slag increased with the increase in frequency at 55 °C. At 10 Hz, the CSM of modified bitumen reference sample, C_3_S-modified bitumen mixture, C_2_S-modified bitumen mixture, and RO phase- modified bitumen reached 1.32 × 10^4^, 3.13 × 10^5^, 2.88 × 10^4^, and 2.74 × 10^4^ Pa, respectively. The CSM values of modified bitumen and the mixtures of bitumen–mineral components of steel slag were decreased in the following order: C_3_S-modified bitumen mixture > C_2_S-modified bitumen mixture > RO phase-modified bitumen mixture > modified bitumen. Results suggested that, at 55 °C, the addition of three mineral phases greatly improved the CSM of the bitumen mixture, and the reinforcement effect of C_3_S was the best.

Figure 11 shows the complex shear modulus of modified bitumen and the mixtures of bitumen–mineral components of steel slag (C_2_S, C_3_S, and RO) in the frequency range of 0.1 to 10 Hz at 45 and 55°C. The complex shear modulus of the mixtures of bitumen–mineral components of steel slag decreased with the increase in temperature.

#### 3.3.3. Rheological Properties of Mineral Phase–Bitumen Mixture at High Temperatures

The rutting factors of bitumen–mineral mixtures at different temperatures are presented in Figure 12. At 45 °C, the rutting factor of modified bitumen was 25 kPa when frequency was 1.592 Hz (Figure 12a), which met the SHRP requirements. Compared with modified bitumen, the three mineral-phase-modified bitumen mixtures showed improved rutting factor, indicating that the HT performance of the mixtures of bitumen–mineral components of steel slag had been improved to different degrees. The rutting factors of C_3_S-modified bitumen mixture, C_2_S-modified bitumen mixture, and RO-modified bitumen mixture were 2.2, 1.49, and 1.51 times that of modified bitumen, respectively. The rutting factors of three mixtures of bitumen–mineral components of steel slag at 45 °C were decreased in the following order: C_3_S-modified bitumen mixture > C_2_S-modified bitumen mixture > RO phase-modified bitumen mixture. At 55 °C, the rutting factors of modified bitumen, C_3_S-modified bitumen mixture, C_2_S-modified bitumen mixture, and RO-modified bitumen mixture at different frequencies are shown in Figure 12b. At 1.592 Hz, the rutting factor of the modified bitumen was about 2.2 kPa, which met the SHRP requirements.

Compared with modified bitumen, the mixtures of bitumen–mineral components of steel slag showed improved rutting factor, indicating that the HT performance of the mixtures of bitumen–mineral components of steel slag had been improved to different degrees. The rutting factors of C_3_S-modified bitumen mixture, C_2_S-modified bitumen mixture, and RO-modified bitumen mixture were 1.58, 1.55, and 1.58 times that of modified bitumen, respectively. The rutting factors of three mixtures of bitumen–mineral components of steel slag at 55 °C were decreased in the following order: C_3_S-modified bitumen > C_2_S-modified bitumen > RO-modified bitumen.

At 45 and 55 °C, the rutting factors of the mixtures of bitumen–mineral components of steel slag (C_2_S, C_3_S, and RO) in the frequency range from 0.1 to 10 Hz are exhibited in Figure 13. The rutting factors of the three mineral-phase–bitumen mixtures decreased with the increase in temperature.

#### 3.3.4. Fourier-Transform Infrared-Spectrum Analysis

FT-IR results of the mixtures of bitumen–mineral components of steel slag are shown in Figure 14. Most absorption peaks in the FT-IR of dicalcium silicate–bitumen mixture and tricalcium silicate–bitumen mixture were the superposition consequence of the absorption peaks of modified bitumen and mineral components (dicalcium and tricalcium silicate), except for a new absorption peak appearing at 3645 cm^−1^. The new peak was ascribed to SiO–H stretching vibration, indicating that dicalcium and tricalcium silicate reacted with modified bitumen. The absorption peaks in RO phase-modified bitumen mixture were the superposition consequence of modified bitumen absorption peaks and RO phase-absorption peaks, indicating that no chemical reaction occurred between modified bitumen and RO phase.

### 3.4. Morphological Characteristics of Steel-Slag–Modified Bitumen Interface

Figure 15 shows SEM images of steel-slag aggregate, steel-slag–bitumen mixture, steel-slag–bitumen sandwich sample interface, and limestone–bitumen sandwich sample interface.

The surface of steel slag was rough and had a pore-like structure (Figure 15a). Pits and grooves were observed on the PSSBM surface (Figure 15b) because the pore-like structure of steel slag was conducive to the formation of deep embedding between steel slag and bitumen, and improved the adhesion of its interface. In Figure 15c, the light-color part of the interface between steel slag and bitumen is steel slag, and the dark-color part is bitumen. In Figure 15d, the light-color part of the interface between limestone and bitumen is limestone aggregate, and the dark-color part is bitumen. The adsorption states of bitumen on limestone and steel-slag surfaces were different (Figure 15c,d). Limestone was compact and had no pores on its surface. The adsorption of bitumen on the surface of limestone was only limited to surface adsorption. In the interface between steel slag and bitumen, bitumen permeated into steel slag from the pores on the surface of the steel slag to form a layer of steel-slag–bitumen composite phase with the steel slag. This composite phase contained the reaction products of bitumen and steel slag. The steel-slag–bitumen interface involved not only physical adsorption but also chemical action. Therefore, pores on the surface of steel slag, and the chemical reaction between bitumen and steel slag greatly enhanced ASS adsorption. Steel slag had properties of a rough surface and pore structures, and could establish a skeleton-like system with bitumen that could improve the interface strength, water permeability, and HT stability of PSSBM.

## 4. Conclusions

In this paper, metallurgical waste, namely, steel slag, was used as aggregate to prepare high-performance permeable bitumen concrete. FT-IR and DSR were used to study the interface chemical-reaction characteristics between bitumen and the main chemical (SiO_2_, Fe_2_O_3_, Al_2_O_3_, and CaO) and mineral components (C_2_S, C_3_S, and RO phase) of steel slag. The influence of various components of steel slag on the adhesion of bitumen to steel-slag interface was discussed. The morphological characteristics of the interface region of steel-slag–bitumen mixtures were studied by SEM. Conclusions are as below.

Under a large bitumen–aggregate ratio, permeable bitumen mixtures could be prepared with steel slag as the aggregate, and the prepared permeable mixture had good permeability, good water, Marshall, and high-temperature stability, and low volume-expansion rate. Bitumen has a good encapsulation effect on steel slag, which could effectively prevent the leaching of heavy-metal ions.Among the four mixtures of bitumen and chemical components, the CaO–bitumen mixture showed the most significant phase-angle change at 45 and 55 °C, the largest complex shear modulus, and the highest rutting factor, thus significantly improving the bonding and high-temperature performance of the interface. A new absorption peak at 3645 cm^–1^ in the infrared spectrum of CaO–bitumen mixture was ascribed to SiO–H stretching vibration from organic silicon compound.Between the three mixtures of bitumen and mineral components, the C_3_S–bitumen mixture showed the most significant phase-angle change at 45 and 55 °C, the largest complex shear modulus, and the highest rutting factor, thus significantly improving the bonding and high-temperature performance of the interface.Pits, grooves, and other textural structures on the surface of steel-slag particles provided a skeleton-like function for the bitumen–steel-slag aggregate interface and improved the adhesion strength of the bitumen–steel-slag aggregate interface.

## Figures and Tables

**Figure 1 materials-13-03885-f001:**
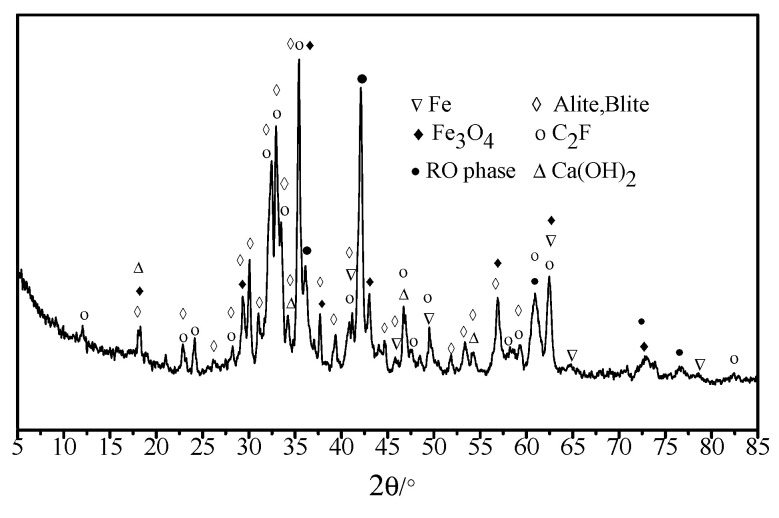
XRD test results of steel-slag aggregate.

**Figure 2 materials-13-03885-f002:**
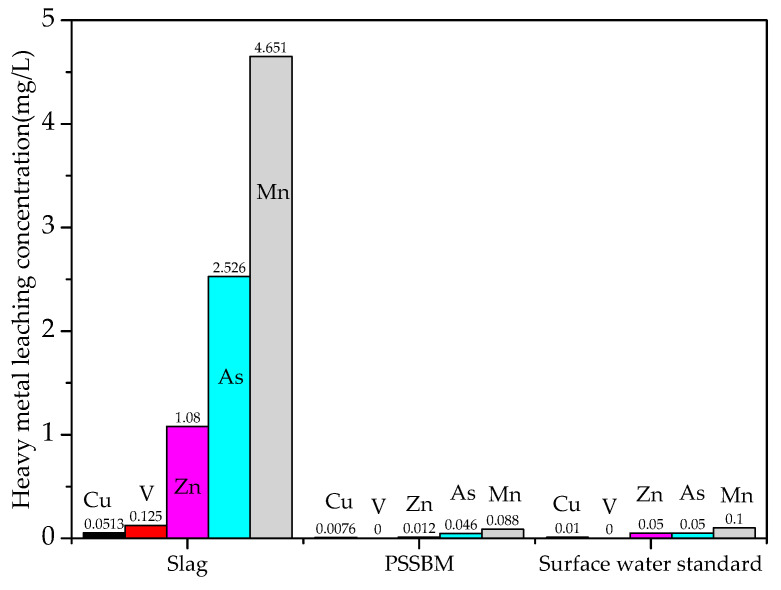
Toxic leaching of steel slag and PSSBM.

**Figure 3 materials-13-03885-f003:**
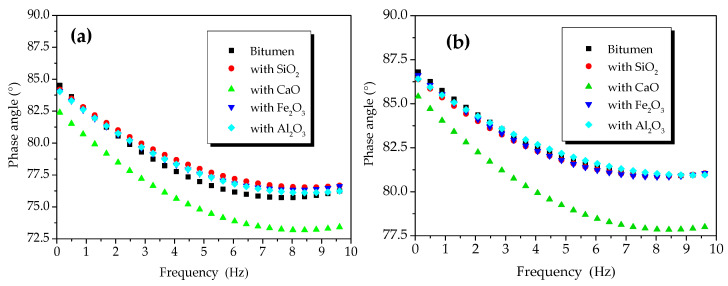
Phase angles of oxide component-modified bitumen mixtures at different temperatures: Phase angle at (**a**) 45 °C and (**b**) 55 °C.

**Figure 4 materials-13-03885-f004:**
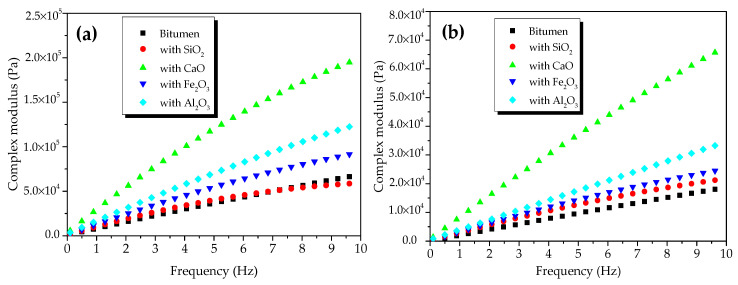
Complex shear modulus of oxide component-modified bitumen mixtures at different temperatures: (**a**) 45 °C and (**b**) 55 °C.

**Figure 5 materials-13-03885-f005:**
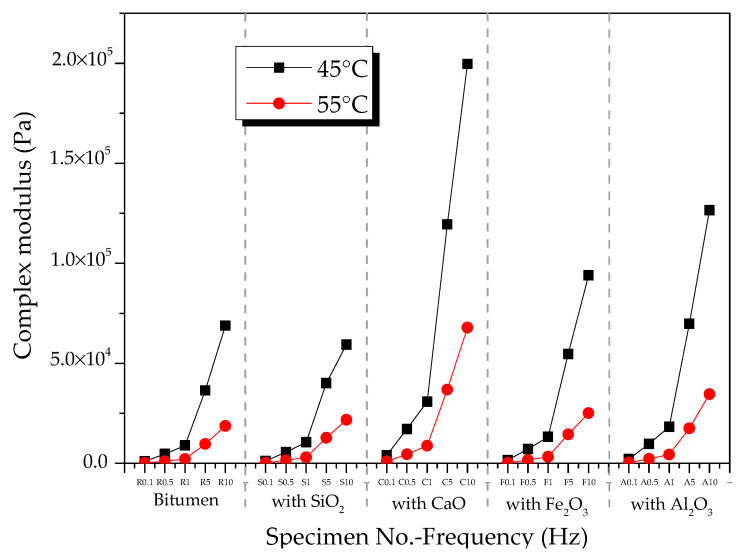
Effects of temperature on complex shear modulus of modified bitumen–oxide mixtures.

**Figure 6 materials-13-03885-f006:**
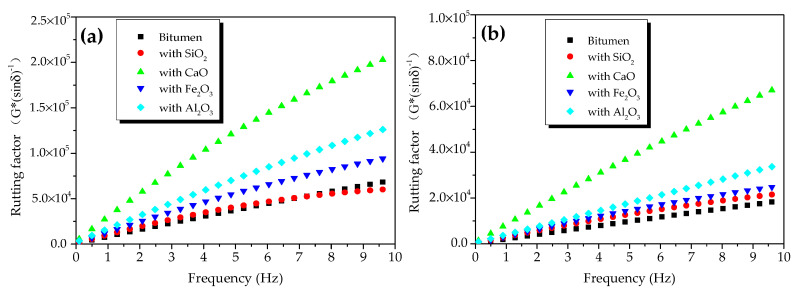
Rutting factors of oxide-modified bitumen mixtures at different temperatures: (**a**) 45 °C and (**b**) 55 °C.

**Figure 7 materials-13-03885-f007:**
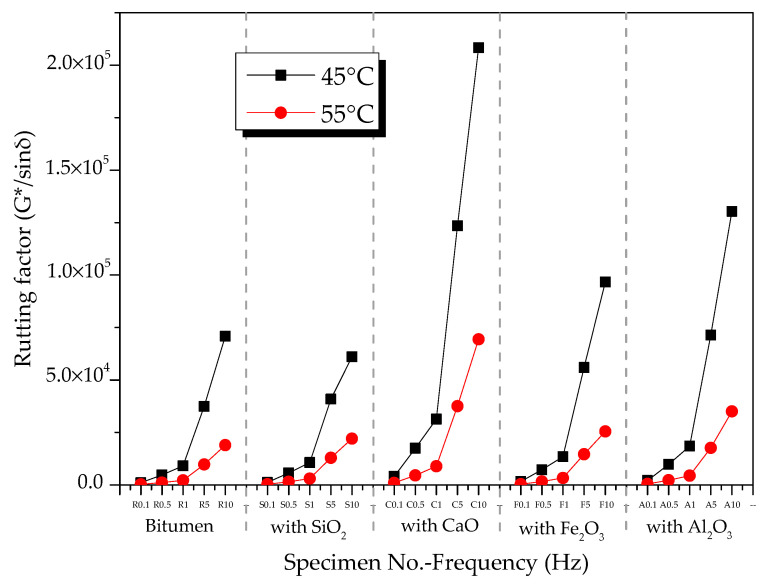
Effects of temperature on rutting factors of oxide-modified bitumen mixtures.

**Figure 8 materials-13-03885-f008:**
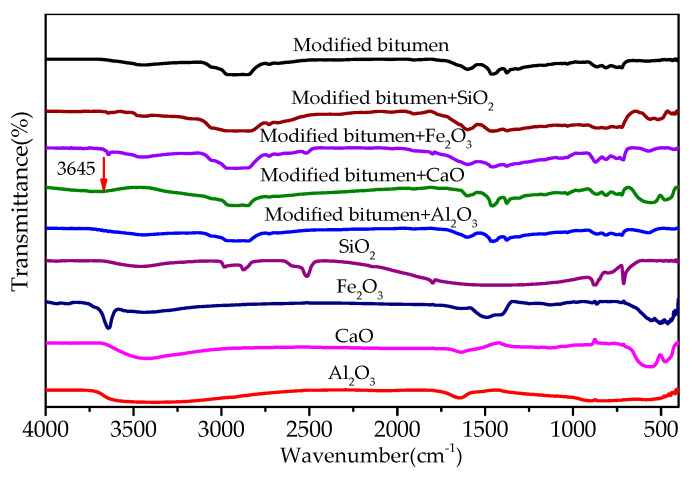
FT-IR results of mixtures of modified bitumen and chemical components of steel slag.

**Figure 9 materials-13-03885-f009:**
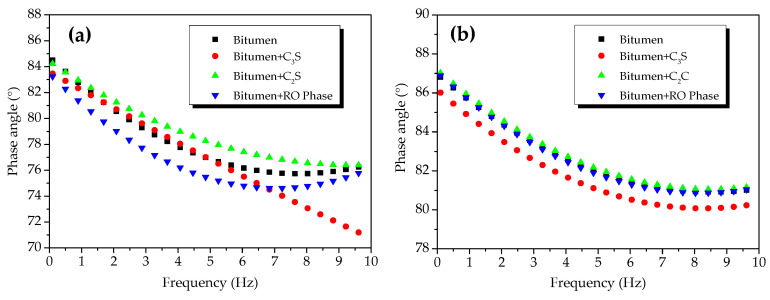
Phase angles of mixtures of modified bitumen and mineral components of steel slag (C_2_S, C_3_S, and RO) at different temperatures: (**a**) 45 °C and (**b**) 55 °C.

**Figure 10 materials-13-03885-f010:**
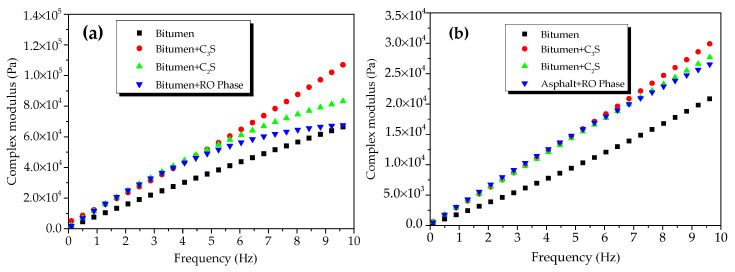
Complex shear modulus of mixtures of bitumen–mineral components of steel slag at different temperatures: (***a***) 45 *°C* and (***b***) 55 °C.

**Figure 11 materials-13-03885-f011:**
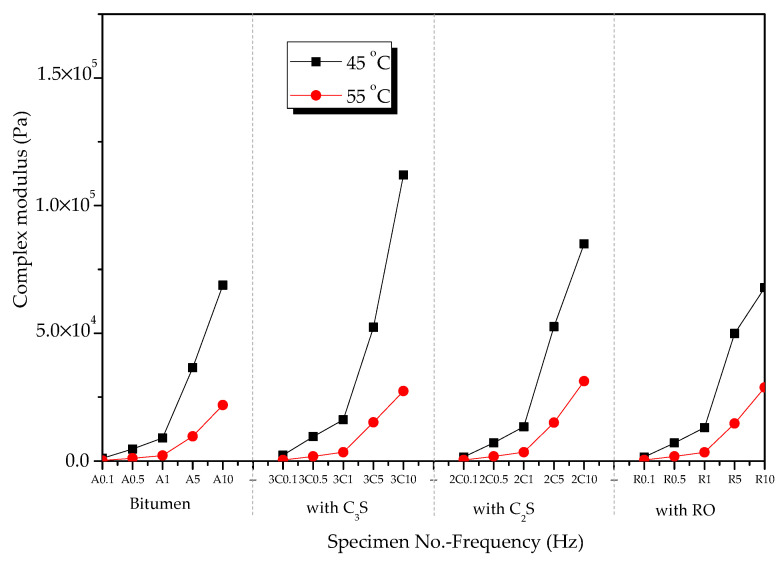
Complex shear modulus of mixtures of bitumen–mineral components of steel slag at different temperatures.

**Figure 12 materials-13-03885-f012:**
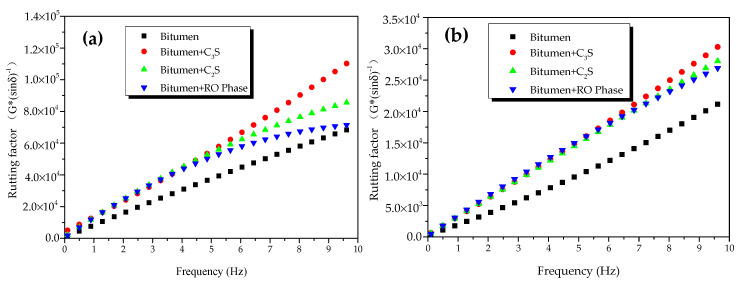
Rutting factors of bitumen–mineral mixtures at different temperatures: (**a**) 45 °C and (**b**) 55 °C.

**Figure 13 materials-13-03885-f013:**
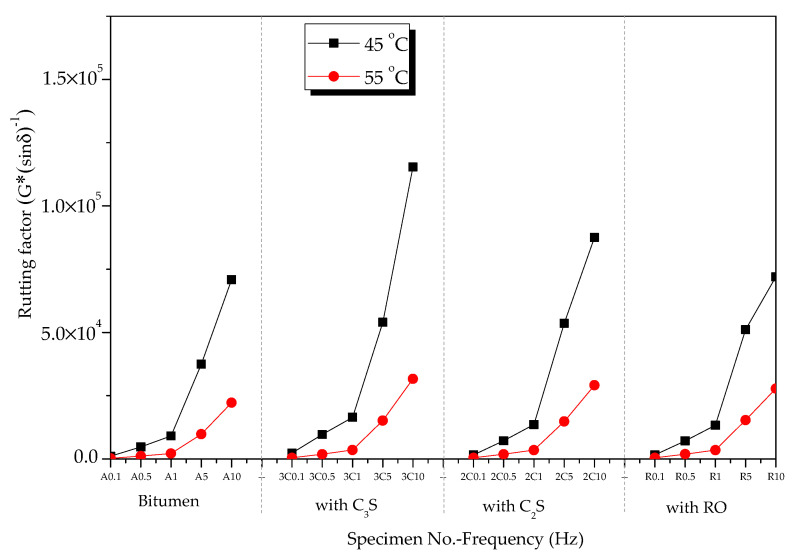
Effect of temperature on rutting factors of mineral phase–bitumen mixtures.

**Figure 14 materials-13-03885-f014:**
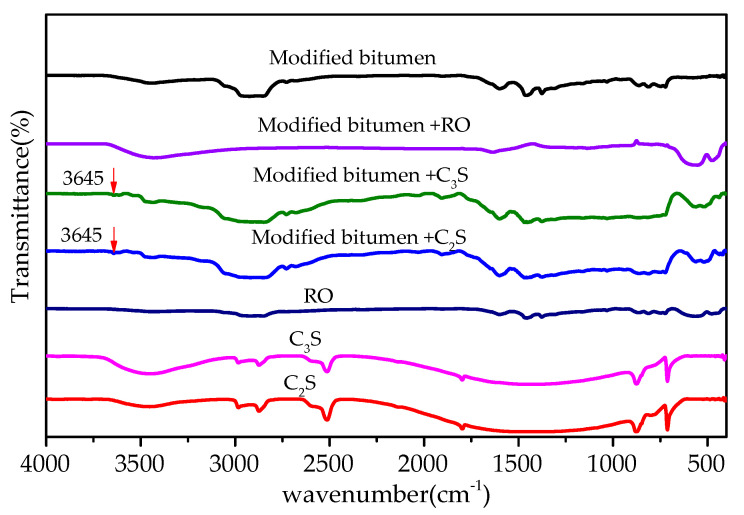
FT-IR of mixtures of bitumen–mineral components of steel slag.

**Figure 15 materials-13-03885-f015:**
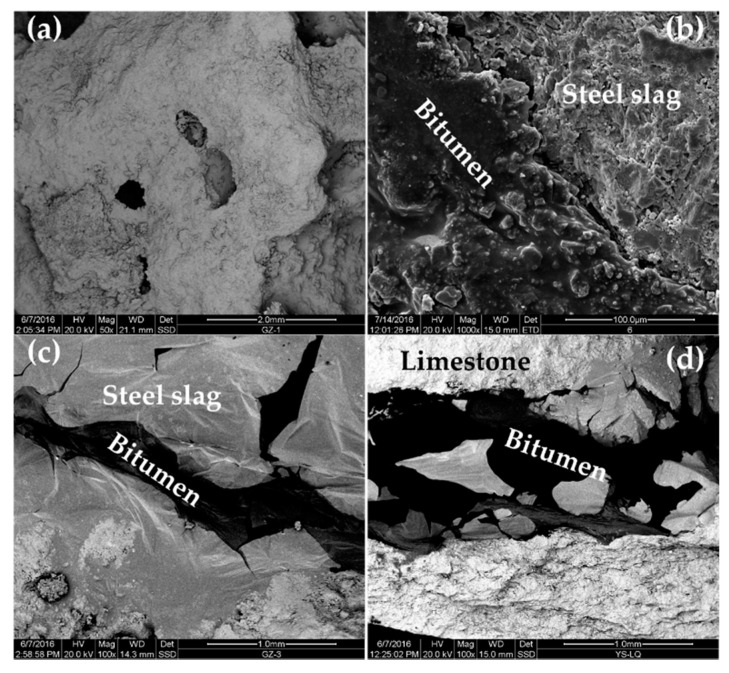
SEM images: (**a**) steel-slag aggregate; (**b**) permeable steel-slag–bitumen mixture; (**c**) steel-slag–bitumen sandwich interface; and (**d**) limestone–bitumen sandwich interface.

**Table 1 materials-13-03885-t001:** Chemical composition of metallurgical steel slag (wt.%).

CaO	Fe_2_O_3_	SiO_2_	Al_2_O_3_	MgO	MnO	P_2_O_5_	TiO_2_	SO_3_	Na_2_O	K_2_O	Others
44.83	21.65	14.38	5.48	3.42	1.94	0.83	0.57	0.23	0.05	0.04	6.58

**Table 2 materials-13-03885-t002:** Technical indices of coarse aggregates.

Test Item	Steel Slag	Request	Normative Test References
Incorruptibility (%)	2.60	≤8.00	CJJ/T 190–2012
Saturated surface-dry density (g/cm^3^)	3.39	≥2.60	CJJ/T 190–2012
Los Angeles wear loss (%)	12.20	≤28.00	CJJ/T 190–2012
Crushing value (%)	11.50	≤26.00	CJJ/T 190–2012
Polished stone value (%)	56.70	≥38.00	CJJ/T 190–2012
Water absorption (by mass %)	1.83	≤2.00	CJJ/T 190–2012

**Table 3 materials-13-03885-t003:** Main technical bitumen indicators.

Test Item	Measured Value	Request	Normative Test References
Penetration (25 °C,100 g, 5 s; 0.1 mm)	50.2	≥40	CJJ/T 190–2012
Ductility (5 cm/min, 5 °C; cm)	42	≥30	CJJ/T 190–2012
Softening point	85	≥80	CJJ/T 190–2012
Flash point (Cleveland open cup (COC); °C)	275	≥260	CJJ/T 190–2012
Dynamic viscosity 60 °C (Pa·s)	32,000	≥20,000	CJJ/T 190–2012

**Table 4 materials-13-03885-t004:** Road performance of permeable steel-slag–bitumen mixture (PSSBM).

Test Item	Measured Value	Specified Value	Reference Standard
Coefficient of permeability (mL/s)	55.56	≥53.33	CJJ/T 190-2012
Void ratio (%)	19.64	18~25	CJJ/T 190-2012
Connected voidage (%)	16.64	≥14	CJJ/T 190-2012
Marshall stability (KN)	9.41	≥5	CJJ/T 190-2012
Residual Marshall stability (%)	91.18	>70	CJJ/T 190-2012
Flow value (mm)	2.56	2~4	CJJ/T 190-2012
Drainage test loss (%)	0.1	<0.3	CJJ/T 190-2012
Cantabro test loss (%)	10.8	<15	CJJ/T 190-2012
Dynamic stability (60 °C, 1 h, times/mm)	6486	≥3500	CJJ/T 190-2012
Intensity ratio of frozen and melted (%)	90	≥85	CJJ/T 190-2012
Volume expansion rate (%)	0.45	≤2.0	CJJ/T 190-2012

**Table 5 materials-13-03885-t005:** Toxic leaching of steel slag and PSSBM.

Metal Leaching Concentration (mg/L)	Cu	V	Zn	As	Mn	Cr
Slag	0.0513	0.125	1.08	2.526	4.651	-
PSSBM	0.0076	-	0.012	0.046	0.088	-
Surface-water standard	≤0.01	-	≤0.05	≤0.05	≤0.1	≤0.01

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
