# Peer review of "Effects of Steel-Slag Components on Interfacial-Reaction Characteristics of Permeable Steel-Slag–Bitumen Mixture"

_materials, 2020, doi:10.3390/ma13173885_

Round 1

Reviewer 1 Report

The study proposes the use of steel slag as a replacement to natural aggregate in order to prepare high-performance asphalt permeable concrete. A range of laboratory tests were conducted to analyse its behaviour.

A few minor changes are necessary for recommending publication of this manuscript.

  1. Abstract is too long. Try curtailing it to include only major outcomes of the study.
  2. References to manufacturers or suppliers can be avoided and instead authors can consider including them in acknowledgement section, if in-kind support is provided.
  3. Refer the standards listed in Table 2 citing relevant references to test specifications/codes. 
  4. Ruggedness, bulk volume relative density, flying loss are not appropriate terms. Check the conventional terms often used in the literature and use them in this manuscript rather than rephrasing.

Reviewer 2 Report

Following article is quite interesting and concerns actual problem of using waste materials (in this case - steel slag) in road construction. However, there are several issues that must be explained and corrected:

1) The authors should use uniform naming in terms of bitumen and asphalt mixtures. The word "asphalt" which is used by authors to describe only (I suppose) bitumen, in American English is used in fact to describe whole asphalt mixture (aggregate + bitumen). Proper naming of bitumen in AmE is "asphalt cement". If British English is being used, word "binder" or "bitumen" is appropriate. This is very important because reading of the present version of the text leads to confusion in many places - e.g. line 188 "oxide-asphalt mixtures" concerns only bitumen but at the first glance it looks like asphalt mixture (aggregate + bitumen) with oxide. Such mistakes should be thoroughly checked and corrected throughout entire text.

2) The authors should be more descritpive about asphalt mixture used in their study. Is it possible to add gradation curve? What type of mixture was used - I suppose that it was some kind of porous asphalt, but it should be clearly noted. Now there is only a designation "permeable asphalt mixture", but there are several types of asphalt mixtures that are permeable. It is hard to compare presented results with others without this information.

3) Initial properties of bitumen should be presented (penetration, softening point, PG, etc.). Now there is only a designation "A-90 road petroleum asphalt" which is quite enigmatic.

4) What was the purpose of describing rubber asphalt in lines 108-113? I could not find any reference to it in the further part of the text. Was it really used in the study?

5) There are some inconstiences between table 4 and figure 2 - according to table 4, V element was detected in slag, but it is not shown in figure 2.

6) I am not suprised that bitumen with CaO achieved best results. Positive effect of adding lime to asphalt mixture (in a slightly different form) is a well known fact for many years. The authors should add a few more details concerning form of different oxides used in their study (e.g. gradation size), because some effects that they observed may have been caused simply because of the "filler effect" of the added oxide in the bitumen.

7) There are grammar errors in many places throughout the text - it must be corrected.

Reviewer 3 Report

This paper studies the influence of using steel slag on the interfacial
reaction characteristics  PSSAM. The topic is interesting but the following comments must be addressed before it can be further considered for publication:

1. The authors are encouraged to study and add the following reference to the introduction:

  • Saghafi, M., Tabatabaee, N., & Nazarian, S. (2019). Performance evaluation of slurry seals containing reclaimed asphalt pavement. Transportation Research Record2673(1), 358-368.
  •  
  • Maslehuddin, M., Sharif, A. M., Shameem, M., Ibrahim, M., & Barry, M. S. (2003). Comparison of properties of steel slag and crushed limestone aggregate concretes. Construction and building materials17(2), 105-112.

2. Several rheological asphalt binders, mastic of binder and slag mineral, and AC mixture test were conducted in this research. It is required (if available) to present the gradation and the volumetric properties of the of AC mixture such as VMA, Gmm, … .

3. How did you come up with the PSV value limit of 38%? In other words, you need to clarify the application of the presented PSSAM (e.g., for highway, local road, etc.) to justify the specified minimum PSV value.

4. Aging and stripping are among the main issues of the porous asphalt mixtures that contain recycle materials. Cantabro Test is one of the main tests that is conducted on the aged and unaged asphalt porous samples for mix design purpose and to evaluate the long-term pavement performance. Please explain why you disregarded this test and how the other tests that you conducted can be considered as a replacement of the Cantabro Test. Also, you did not elaborate on the stripping potential of the PSSAM.
Please explain why you omitted the stripping potential of the mixture while it is an essential performance property of porous mixture.

5. DSR tests are usually conducted in the short-term (rutting) and long-term (cracking) aged samples. Please explain how you came up with the current DSR test criteria?

6. An overall English revision is recommended.

Reviewer 4 Report

This paper describes the rheological outcomes of blending a range of mineral oxide materials with asphalt (bitumen), with the aim of unpicking the underlying driver of the effect of blending waste steel-making slag with bitumen. 

The authors suggest that they have performed a range of instrumental and chemical analyses, but some of these are not included in the paper, eg FTIR.

The leaching results are interesting and relevant, and indicate the difference between natural and bitumen-coated slag particles. The authors do not however indicate what the level of leaching will be when the coated particles become un-coated by traffic wear in an asphalt surfacing (i.e. wear, not stripping).

The authors quite clearly demonstrate that the CaO or C3S phases have the largest effect on the rheology of the blends and correlate that with the performance of the multi-component steel slag material. The surface morphology of the steel slag is also considered important.

While the authors have shown many outcome measurements they have not identified the direct interactions that lead to those outcomes, they have rather inferred them, based on the deep body of work in bitumen adhesion that is available.

My opinion is that the paper is missing a section where the authors will investigate the chemical interactions between the slag components and bitumen binder. 

Round 2

Reviewer 2 Report

Thank You for providing revised version of the original manuscript. In my opinion now it can be accepted in the present form.

Author Response

Dear professor, this is my reply to your Comments and Suggestions and I hope it can meet your requirements.

1.Thank You for providing revised version of the original manuscript. In my opinion now it can be accepted in the present form.

Answer: Thank you for your support and encouragement to the author. Thank you very much for your valuable amendments to the manuscript.

Reviewer 4 Report

The revised paper appears to have been well revised.

One or two small points have been missed:

PSSAM as used in line 15. This should be changed to PSSBM as used elsewhere.

RO phase is not defined. This should be added.

The leaching results and conclusions are very nice and I think provide a good scheme for other workers looking at other materials.

However, I have issue with the assignment of the very weak and very broad IR band around 3645 wavenumbers. The authors suggest it to be due to SiO-H stretching mode possibly from an organo-silicon species. However, this band was observed in the bitumen/CaO mixture. Therefore, it would appear that the band cannot be assigned to SiO-H stretching mode. The weakness of the band indicates low population of species, and the broadness indicates many different energy states, compatible with Hydrogen-bonded species, not discrete covalent species. Indeed, the spectrum of Ca-O shows the broad H-bonded O-H stretching modes around 3450 cm-1. Interaction with bitumen might easily cause shifts to higher wavenumbers and reduce their intensities. Overall, in my opinion this band is simply indicative of broad and weak H-bonding between surface OH groups on the CaO and the bitumen matrix and only moderately interesting.

Author Response

Dear professor, this is my reply to your Comments and Suggestions and I hope it can meet your requirements.

1. PSSAM as used in line 15. This should be changed to PSSBM as used elsewhere.

Answer: Thank you for your valuable suggestions. The author has corrected the mistakes in the manuscript, and has corrected PSSAM to PSSBM in line 15.

2. RO phase is not defined. This should be added.

Answer: Thank you for your valuable suggestions. The author has added the definition of RO phase in line 101 of the manuscript, as shown below: “RO phase (RO phase is a broad solid solution formed by melting FeO, MgO, and other divalent metal oxides such as MnO)”.

3. However, I have issue with the assignment of the very weak and very broad IR band around 3645 wavenumbers. The authors suggest it to be due to SiO-H stretching mode possibly from an organo-silicon species. However, this band was observed in the bitumen/CaO mixture. Therefore, it would appear that the band cannot be assigned to SiO-H stretching mode. The weakness of the band indicates low population of species, and the broadness indicates many different energy states, compatible with Hydrogen-bonded species, not discrete covalent species. Indeed, the spectrum of Ca-O shows the broad H-bonded O-H stretching modes around 3450 cm-1. Interaction with bitumen might easily cause shifts to higher wavenumbers and reduce their intensities. Overall, in my opinion this band is simply indicative of broad and weak H-bonding between surface OH groups on the CaO and the bitumen matrix and only moderately interesting.

Answer:  Thank you for your valuable suggestions. Because bitumen contains no water, the author thinks that calcium oxide will not produce calcium ions and hydroxide ions in bitumen, so we infer that calcium oxide does react with the modified bitumen around 3645 wavenumbers and form a new substance, which may have been an organosilicon compound, as inferred from the characteristic peak.